# AAV2.7m8 is a powerful viral vector for inner ear gene therapy

Kevin Isgrig[1], Devin S. McDougald[2], Jianliang Zhu[1], Hong Jun Wang[1], Jean Bennett[2] & Wade W. Chien[1,3]

Adeno-associated virus (AAV) has been successfully used to deliver gene therapy to improve auditory function in mouse models of hereditary hearing loss. Many forms of hereditary hearing loss have mutations which affect the cochlear hair cells, the mechanosensory cells which allow for sound detection and processing. While most conventional AAVs infect inner hair cells (IHCs) with various efficiencies, they infect outer hair cells (OHCs) and supporting cells at lower levels in the cochlea. Here we examine the infection patterns of two synthetic AAVs (AAV2.7m8 and AAV8BP2) in the mouse inner ear. AAV2.7m8 infects both IHCs and OHCs with high efficiency. In addition, AAV2.7m8 infects inner pillar cells and inner phalangeal cells with high efficiency. Our results suggest that AAV2.7m8 is an excellent viral vector for inner ear gene therapy targeting cochlear hair cells and supporting cells, and it will likely greatly expand the potential applications for inner ear gene therapy.

[1] Neurotology Program, National Institute on Deafness and Other Communication Disorders (NIDCD), National Institutes of Health, Bethesda, MD 20892, USA. [2] Center for Advanced Retinal and Ocular Therapeutics, Perelman School of Medicine, University of Pennsylvania, Philadelphia, PA 19104, USA. [3] Department of Otolaryngology-Head and Neck Surgery, Johns Hopkins School of Medicine, Baltimore, MD 21287, USA. Correspondence and requests for materials should be addressed to W.W.C. (email: wade.chien@nih.gov)

Hearing loss is one of the most common disabilities affecting the world's population today. According to the National Health and Nutritional Examination Survey, nearly two thirds of US adults aged 70 years and older are affected by hearing loss[1]. The mammalian cochlea contains two types of hair cells, inner hair cells (IHCs) and outer hair cells (OHCs), both of which are important for the detection and processing of auditory information[2]. These hair cells are surrounded by supporting cells, a heterogeneous group of cells that are important for cochlear homeostasis[3]. The mature mammalian hair cells are incapable of regeneration[4]. Therefore, once the damage occurs in these cells, the degeneration process is often irreversible.

Inner ear gene therapy is a promising therapeutic modality that can potentially prevent and reverse hair cell damage[5]. Several studies have shown that viral vector-mediated inner ear gene therapy can be applied to animal models of hereditary hearing loss to improve auditory function[6–12]. The majority of these studies used adeno-associated virus (AAV) for gene delivery. AAV is a single-stranded DNA parvovirus[5]. It is a commonly used viral vector in human gene therapy clinical trials due to the fact that it is non-pathogenic in humans[5]. While several AAV serotypes have been shown to infect IHCs effectively, OHC infection rates have been low[7,9]. In addition, the infection efficiency of conventional AAVs for cochlear supporting cells is also low[13,14]. In order for the inner ear gene therapy to achieve complete hearing restoration, a viral vector with higher infection efficiency is required.

Various strategies have been used to enhance the infection efficiency and specificity of AAVs; these efforts have led to the production of synthetic AAVs that have superior infection efficiencies[15]. Two of the novel synthetic AAVs that have been shown to have enhanced cellular transduction in the retina are AAV2.7m8 and AAV8BP2[16,17]. AAV2.7m8 was generated using an in vivo-directed evolution approach where AAV libraries with diverse capsid protein modifications were screened for the infection efficiency of mouse photoreceptor cells via intravitreal injection[16]. This vector contains a 10-amino acid peptide inserted at position 588 of the AAV2 capsid protein sequence, which is involved with AAV2 binding to its primary receptor, heparan sulfate proteoglycan[16,18]. Similarly, AAV8BP2 was generated using an in vivo-directed evolution approach in which AAV libraries were screened for the infection of mouse retinal bipolar cells via subretinal injection. This vector contains modifications at amino acids 585–594 of the AAV8 capsid protein sequence[17].

In this study, we examine the infection patterns of AAV2.7m8 and AAV8BP2 in the mouse inner ear. We show that AAV2.7m8 is capable of infecting the cochlear IHCs and OHCs with high efficiency. We also show that AAV2.7m8 is capable of infecting the inner pillar cells and inner phalangeal cells with high efficiency. These results suggest that AAV2.7m8 is a powerful viral vector for inner ear gene delivery.

## Results

**AAV2.7m8 infects cochlear hair cells with high efficiency.** To assess the infection efficiency of synthetic AAVs in the mammalian inner ear, AAV2.7m8-GFP ($9.75 \times 10^{12}$ genome copies [GC]/mL) and AAV8BP2-GFP ($1.10 \times 10^{13}$ GC/mL) were delivered to neonatal (P0–P5) mouse inner ears using the posterior semicircular canal approach. Posterior semicircular canal gene delivery allows viral vectors to effectively infect cells in the neonatal cochlea as well as vestibular organs[7,14,19]. Infection efficiencies of AAV2-GFP ($5.69 \times 10^{12}$ GC/mL) and AAV8-GFP ($1.66 \times 10^{13}$ GC/mL), the two commonly used conventional AAVs from which AAV2.7m8 and AAV8BP2 are derived respectively, as well as the synthetic AAV Anc80L65-GFP ($1.89 \times$

$10^{13}$ GC/mL), were also examined using the same delivery approach as additional controls. One microliter of AAV was delivered into each animal. Hair cell infection efficiency was assessed by quantifying the percentage of hair cells (identified by anti-Myo7a antibody) with green fluorescent protein (GFP) expression. Examination of the cochlea 4 weeks after gene delivery revealed high levels of GFP in both IHCs and OHCs in mice that were injected with AAV2.7m8-GFP ($n = 8$ animals; Fig. 1, Supplementary Table 1). The overall infection efficiency was $84.1 \pm 5.66\%$ (mean ± standard error) for IHC and $83.1 \pm 6.17\%$ for OHC. Mice injected with AAV8BP2-GFP ($n = 9$ animals; Fig. 1, Supplementary Table 1) had moderate-to-high levels of GFP expression in IHCs and OHCs. The overall infection efficiency was $55.7 \pm 9.53\%$ for IHC and $44.1 \pm 7.94\%$ for OHC ($p = 0.016$ and $< 0.001$ [$t$ test] for IHC and OHC respectively, when compared to AAV2.7m8).

Comparison of AAV2.7m8-GFP to conventional AAVs also showed superior cochlear hair cell infection efficiency, particularly with regard to OHCs. For AAV2-GFP ($n = 3$ animals; Fig. 1, Supplementary Table 1), the overall infection efficiency was $43.6 \pm 13.5\%$ for IHC and $54.5 \pm 12.7\%$ for OHC ($p = 0.003$ and $0.03$ [$t$ test] for IHC and OHC respectively, when compared to AAV2.7m8). For AAV8-GFP ($n = 4$ animals; Fig. 1, Supplementary Table 1), the overall infection efficiency was $86.0 \pm 5.34\%$ for IHC and $51.7 \pm 5.95\%$ for OHC ($p = 0.84$ and $0.003$ [$t$ test] for IHC and OHC respectively, when compared to AAV2.7m8).

Anc80L65 is a synthetic AAV that has been reported to infect both IHCs and OHCs with high efficiency[20]. When Anc80L65-GFP was injected into neonatal mouse inner ears using the posterior canal approach ($n = 7$ animals; Fig. 1, Supplementary Table 1), the overall infection efficiency was $94.0 \pm 3.63\%$ for IHC and $67.0 \pm 4.32\%$ for OHC. While the IHC infection efficiency is comparable between AAV2.7m8 and Anc80L65 ($p = 0.16$), our data suggest that AAV2.7m8 is more capable of infecting OHCs compared to Anc80L65 ($p = 0.04$, $t$ test).

A detailed examination of neonatal mice injected with AAV2.7m8-GFP ($n = 8$ animals) showed that AAV2.7m8 was able to infect hair cells throughout the entire cochlea (Fig. 2). The IHC infection efficiency was $90.3 \pm 8.98\%$ at the cochlear apex, $84.6 \pm 10.4\%$ at the middle turn, and $77.5 \pm 10.8\%$ at the cochlear base. The OHC infection efficiency was $89.0 \pm 9.53\%$ at the cochlear apex, $85.2 \pm 10.9\%$ at the middle turn, and $74.9 \pm 12.2\%$ at the cochlear base. In four out of the eight mice that were injected with AAV2.7m8, the IHC and OHC infection rates were over 90% throughout the entire cochlea (Fig. 2). Delivery of AAV2.7m8-GFP to adult CBA/J mice (1- to 6-month-old, $n = 6$ animals) showed that it is equally competent at infecting the IHCs (overall infection efficiency $84.5 \pm 4.91\%$) and OHCs (overall infection efficiency $74.9 \pm 6.53\%$) in the adult cochlea (Supplementary Fig. 1). Taken together, our results indicate that AAV2.7m8 is a powerful viral vector that is capable of infecting both cochlear IHCs and OHCs with high efficiency.

**AAV2.7m8 infects vestibular hair cells less efficiently.** In addition to assessing the hair cell infection efficiency of synthetic AAVs in the cochlea, the hair cell infection efficiency was also examined in the vestibular organs. When AAV2.7m8-GFP and AAV8BP2-GFP were delivered to neonatal mouse inner ears, GFP was expressed in vestibular organs. Quantification of vestibular hair cell infection efficiency was done in the utricle (Fig. 3, Supplementary Table 1). The utricular hair cell infection efficiency was $27.5 \pm 9.65\%$ for AAV2.7m8-GFP ($n = 8$ animals) and $34.2 \pm 9.84\%$ for AAV8BP2-GFP ($n = 9$ animals, $p = 0.63$ [$t$ test] compared to AAV2.7m8). The vestibular hair cell infection efficiency of AAV2-GFP, AAV8-GFP, and Anc80L65-GFP was also

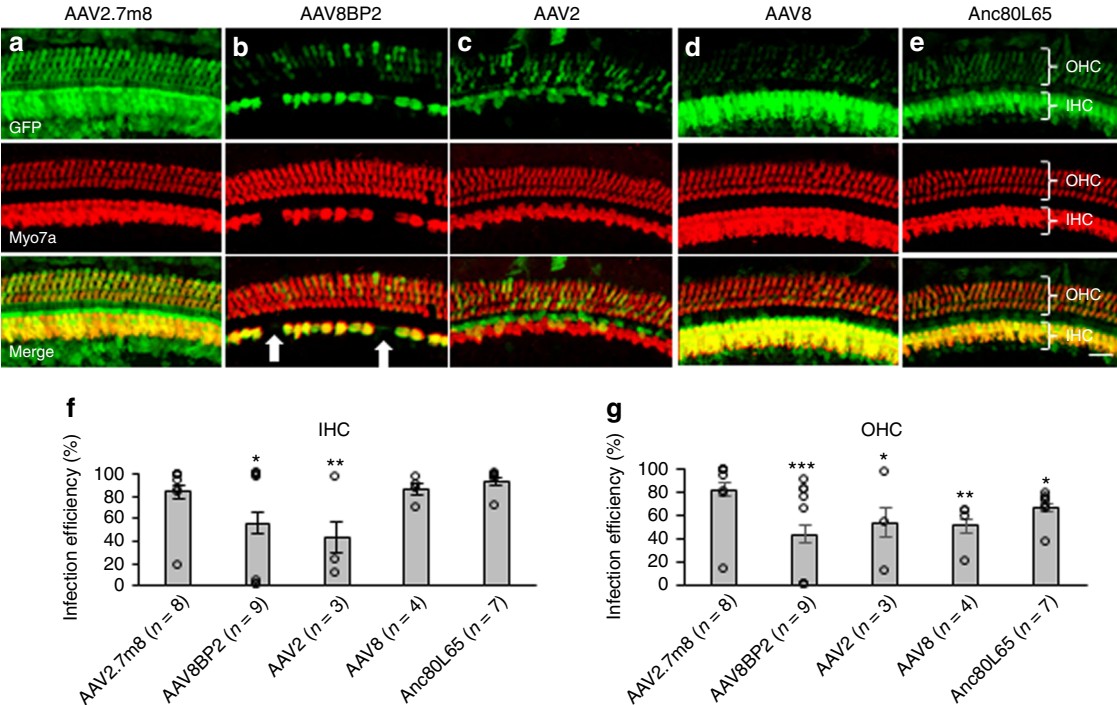

**Fig. 1** AAV2.7m8 infects cochlear inner and outer hair cells with high efficiency. **a–e** When AAV2.7m8-GFP (**a**) was injected into neonatal mouse inner ear via the posterior semicircular canal approach, the IHCs and OHCs were infected with high efficiency throughout the entire cochlea. AAV8BP2-GFP (**b**) injection caused some loss in IHCs (white arrows). AAV2-GFP (**c**), AAV8-GFP (**d**), and Anc80L65-GFP (**e**) infected IHCs at high levels, but the OHC infection efficiency was less than AAV2.7m8-GFP. GFP expression is shown in green and Myo7a expression (a marker for hair cells) is shown in red. 40x images of the cochlear apex are shown. Scale bar represents 20μm. **f, g** Quantification of IHC (**f**) and OHC (**g**) infection efficiency. Open circles represent the average infection efficiency of each animal. $n$ represents the number of animals tested. For each animal, hair cell infection was quantified at six different locations along the cochlea: two at the apex, two at the middle turn, and two at the cochlear base. Error bars represent standard errors. Statistical significance with reference to AAV2.7m8 is shown above error bars (* represents $p < 0.05$, ** represents $p < 0.01$, and *** represents $p < 0.001$; $t$-test). Source data are provided as a Source Data file. IHC, inner hair cell; OHC, outer hair cell; $p$, $p$ values

examined in neonatal mouse utricles in vivo (Fig. 3, Supplementary Table 1). The utricular hair cell infection efficiency was 32.4 ± 6.16% for AAV2 ($n = 3$ animals, $p = 0.77$ compared to AAV2.7m8), 93.3 ± 2.15% for AAV8 ($n = 4$ animals, $p < .001$ [$t$ test] compared to AAV2.7m8), and 67.7 ± 2.46% for Anc80L65 ($n = 7$ animals, $p = 0.002$ [$t$ test] compared to AAV2.7m8). These results indicate that AAV2.7m8 preferentially infects cochlear hair cells at much higher efficiency than vestibular hair cells.

**AAV2.7m8 infects LGR5+ supporting cells with high efficiency.** While cochlear hair cells have garnered the most attention as the targeted cell type in inner ear gene therapy studies, the glia-like supporting cells that surround hair cells are also important therapeutic targets for gene therapy. A specific subset of supporting cells, namely inner pillar cells, inner phalangeal cells, and the third row of Deiters cells, express Leucine-rich repeat-containing G-protein coupled receptor 5 (LGR5) and demonstrate progenitor cell-like properties that promote hair cell regeneration[21,22]. When AAV2.7m8-GFP was delivered to neonatal mouse inner ears, GFP expression was seen in two of these LGR5 + supporting cell types, inner pillar cells and inner phalangeal cells (Fig. 4, Supplementary Fig. 2, Supplementary Table 1). The overall inner pillar cell infection efficiency was 86.1 ± 4.56% (94.7 ± 3.11% at the apex, 91.3 ± 3.80% at the middle turn, and 72.4 ± 7.93% at the base; $n = 8$ animals). The overall inner phalangeal cell infection efficiency was 61.4 ± 9.30% (72.0 ± 18.4% at the apex, 60.0 ± 16.3% at the middle turn, and 52.3 ± 16.8% at the base; $n = 4$ animals). In contrast, mice injected with AAV8BP2 had no GFP expression in the inner pillar cells and

inner phalangeal cells (Fig. 4). Inner pillar cell infection was also seen in mice injected with AAV2-GFP (60.3 ± 7.96%, $n = 3$ animals, $p = 0.007$ [$t$ test] compared to AAV2.7m8), AAV8-GFP (50.4 ± 7.49%, $n = 4$ animals, $p < 0.001$ [$t$ test] compared to AAV2.7m8), and Anc80L65-GFP (75.3 ± 4.94%, $n = 7$ animals, $p = 0.11$ [$t$ test] compared to AAV2.7m8). However, none of these AAVs infected inner phalangeal cells. These results suggest that AAV2.7m8 is capable of infecting the subset of supporting cells (inner pillar cells and inner phalangeal cells) that are thought to be capable of promoting hair cell regeneration with high efficiency.

**AAV2.7m8 is safe for use in mammalian inner ear.** In order for the inner ear gene therapy to be a viable treatment for hearing loss and vestibular dysfunction, the viral vector used should have minimal effect on normal auditory and vestibular functions. To assess whether the inner ear delivery of synthetic AAVs had any effect on hearing, auditory brainstem responses (ABRs) were measured (Fig. 5). Mice that underwent AAV2.7m8-GFP ($n = 8$ animals), AAV2-GFP ($n = 3$ animals), AAV8-GFP ($n = 4$ animals), and Anc80L65-GFP ($n = 7$ animals) injections showed no significant change in ABR thresholds compared to the control mice that underwent no inner ear manipulation ($p = 0.09$, 0.25, 0.43, and 0.25, respectively, ANOVA). In contrast, mice that underwent AAV8BP2-GFP ($n = 13$ animals) injection showed a 10–25 dB ABR threshold elevation compared to the control mice ($p < .001$, analysis of variance (ANOVA)). Post-hoc comparisons using Scheffe's method showed statistically significant ABR threshold differences at 4, 8, 16, and 32 kHz ($p = 0.004$, < 0.001,

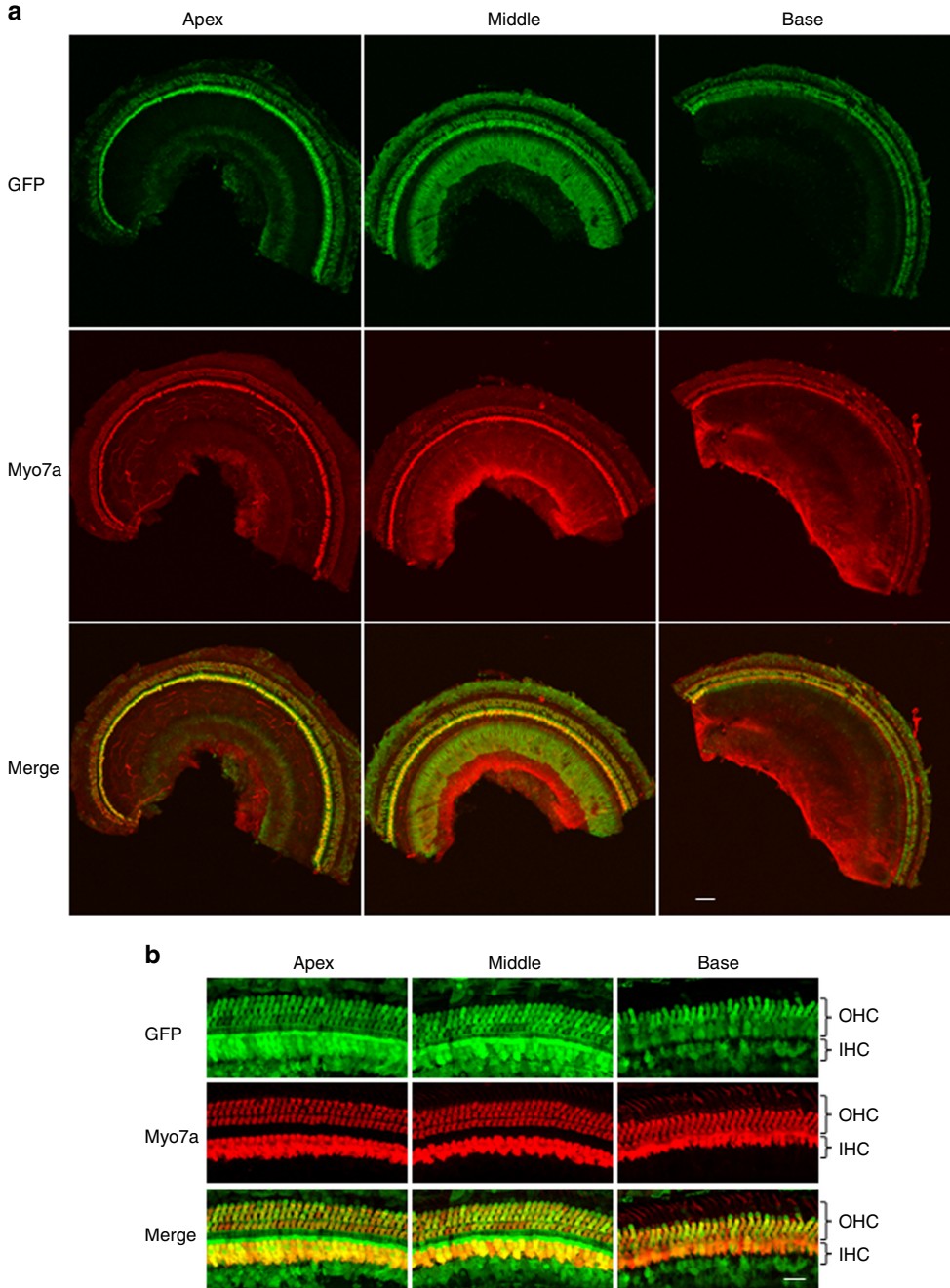

**Fig. 2** AAV2.7m8 infects inner and outer hair cells throughout the entire cochlea.10 × (**a**) and 40 × (**b**) images of a mouse cochlea that underwent AAV2.7m8-GFP injection via the posterior semicircular canal approach. GFP expression is seen in both IHCs and OHCs throughout the entire cochlea. GFP expression is shown in green, and Myo7a expression (a marker for hair cells) is shown in red. Scale bar represents 50 μm for 10 × and 20 μm for 40 × images

< 0.001, and 0.034, respectively). It is possible that AAV8BP2 is more immunogenic to the mouse inner ear, which leads to cochlear hair cell loss (Fig. 1) as well as ABR threshold elevation. Examination of the cochlea after AAV8BP2 injection revealed infiltration of inflammatory cells (Supplementary Fig. 3). When AAV8BP2-GFP was injected at half of the original concentration $(0.55 \times 10^{10}$ GC), the ABR thresholds were comparable to those of control mice ($p = 0.49$, Supplementary Fig. 4), but the IHC and OHC infection efficiency also decreased (43.2 ± 8.36% and 23.3 ± 5.41%, respectively; $n = 5$ animals), though the changes were not statistically significant ($p = 0.38$ and 0.08 [$t$ test] for IHC and OHC, respectively).

Mice with vestibular dysfunction often exhibit circling behavior[23]. To assess whether the inner ear delivery of synthetic AAVs had any effect on the vestibular system, the circling behavior of injected mice was examined (Fig. 5). Control mice that did not undergo inner ear gene delivery circled 5.11 ± 0.32 times per 2 min ($n = 6$ animals). The circling behavior of mice injected with AAV2.7m8-GFP (5.04 ± 0.54 times per 2 min, $n = 8$ animals), AAV2-GFP (6.00 ± 1.02 times per 2 min, $n = 3$ animals), AAV8-GFP (4.58 ± 0.28 times per 2 min, $n = 4$ animals), and Anc80L65-GFP (5.52 ± 0.65 times per 2 min, $n = 7$ animals) was similar to that of non-injected control mice ($p = 0.92, 0.31, 0.28,$ and 0.60, respectively, ANOVA). In contrast,

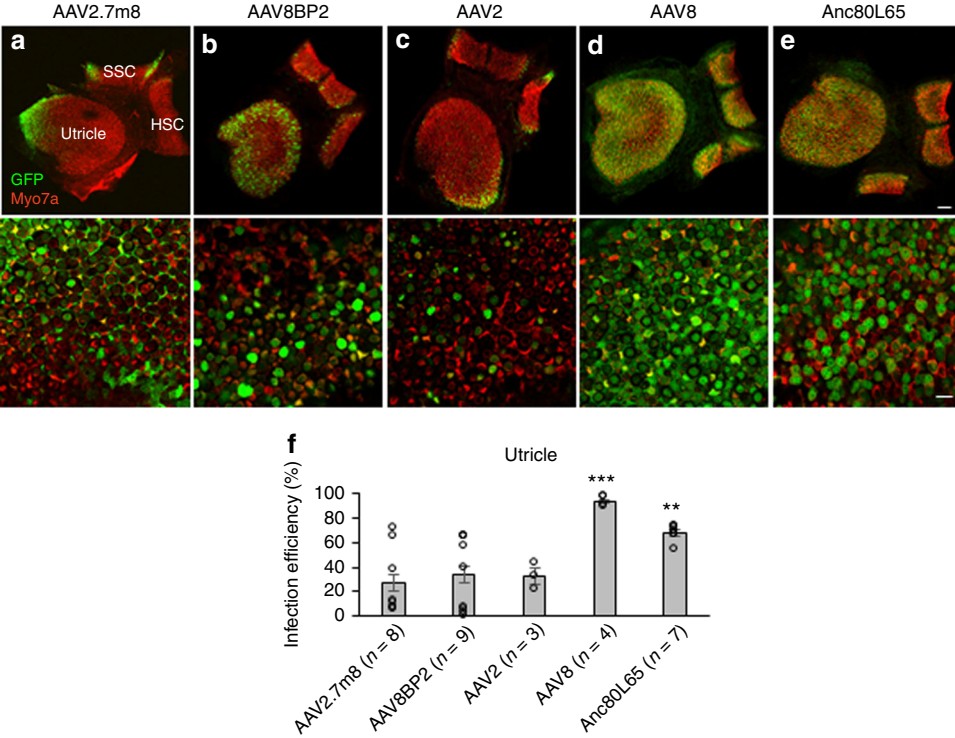

**Fig. 3** AAV2.7m8 infects vestibular hair cells with lower efficiency. **a**–**e** 10 × and 40 × images of utricles showing the hair cell infection efficiency in response to posterior canal AAV delivery. AAV2.7m8-GFP (**a**), AAV8BP2-GFP (**b**), AAV2-GFP (**c**) infected utricular hair cells at lower levels. In contrast, AAV8-GFP (**d**) and Anc80L65-GFP (**e**) infected utricular hair cells at higher levels. GFP expression is shown in green and Myo7a expression (a marker for hair cells) is shown in red. Scale bar represents 50μm for 10 × and 100 μm for 40 × images. **f** Quantification of utricular hair cell infection efficiency. Statistical significance with reference to AAV2.7m8 is shown above error bars (* represents $p < 0.05$, ** represents $p < 0.01$, and *** represents $p < 0.001$; t-test). Error bars represent standard errors. Open circles represent the average infection efficiency of each animal. n represents the number of animals tested. For each animal, hair cell infection was quantified at two different locations in each utricle. Source data are provided as a Source Data file. SSC, superior semicircular canal; HSC, horizontal semicircular canal. p, p values

mice that underwent AAV8BP2-GFP injection had a slight increase in circling ($6.87 \pm 0.38$ times per 2 min, $p = 0.009$, $n = 13$ animals). Injection of AAV8BP2-GFP at half of the original concentration ($0.55 \times 10^{10}$ GC) resulted in no increase in circling behavior compared to control animals ($5.47 \pm 0.77$ times per 2 min, $p = 0.66$, $n = 5$ animals; Supplementary Fig. 4). These results suggest that the inner ear delivery of AAV2.7m8 is safe and resulted in no adverse effect in auditory and vestibular functions.

## Discussion

While several studies have shown that the viral inner ear gene therapy improves auditory function in mouse models of hereditary hearing loss, the hearing recovery is often incomplete[6–9]. One of the main drawbacks of conventional AAVs is that they infect the OHCs with low efficiency. This problem has been overcome with the introduction of the synthetic AAV Anc80L65, which has been shown to infect cochlear IHCs and OHCs with high efficiency[20]. Our results suggest that AAV2.7m8 is also capable of infecting cochlear IHCs and OHCs with high efficiency. In fact, AAV2.7m8 infects OHCs at even higher efficiency than Anc80L65 when delivered through the posterior canal approach. The differences in Anc80L65 OHC infection efficiency between our data and the published work may reflect the different delivery approaches (posterior canal vs round window approach) as well as the different promoters used (CAG vs CMV)[20]. While our data indicate that AAV2.7m8 is highly effective at infecting cochlear hair cells in mouse cochlea, it remains important to confirm these findings in larger animals such as non-human

primates in order for this virus to be useful in human inner ear gene therapy.

We found that AAV2.7m8 preferentially targeted the cochlear hair cells compared to the vestibular hair cells. This is different from Anc80L65, which also infects vestibular hair cells with high efficiency[20]. The predilection of AAV2.7m8 for targeting cochlear hair cells may be useful in studies in which transgene expression is only desirable in the cochlea, and it can potentially minimize the vestibular toxicity from unwanted transgene expression in the vestibular system.

Most inner ear gene therapy studies have focused on animal models of hereditary hearing loss. However, the prevalence of hereditary hearing loss is much lower than the other types of hearing losses, such as age-related hearing loss (presbycusis) and noise-induced hearing loss. One strategy for applying gene therapy to treat presbycusis and noise-induced hearing loss is to induce hair cell regeneration. While the hair cells of non-mammalian animals (such as birds and zebrafish) are regenerated after damage[24], mammalian hair cells are not regenerated. The supporting cells are thought to serve as a source for hair cell regeneration[25]. In mammalian inner ear, the subset of supporting cells that are LGR5 + (inner pillar cells, inner phalangeal cells, and the third row of Deiters cells) have progenitor cell-like properties that promote hair cell regeneration[21]. In order to utilize gene therapy to induce hair cell regeneration, one critical element is to have a viral vector that can effectively target this population of supporting cells. In this study, we showed that AAV2.7m8 effectively infects both IHCs and OHCs in the cochlea. In addition, it also infects the types of supporting cells

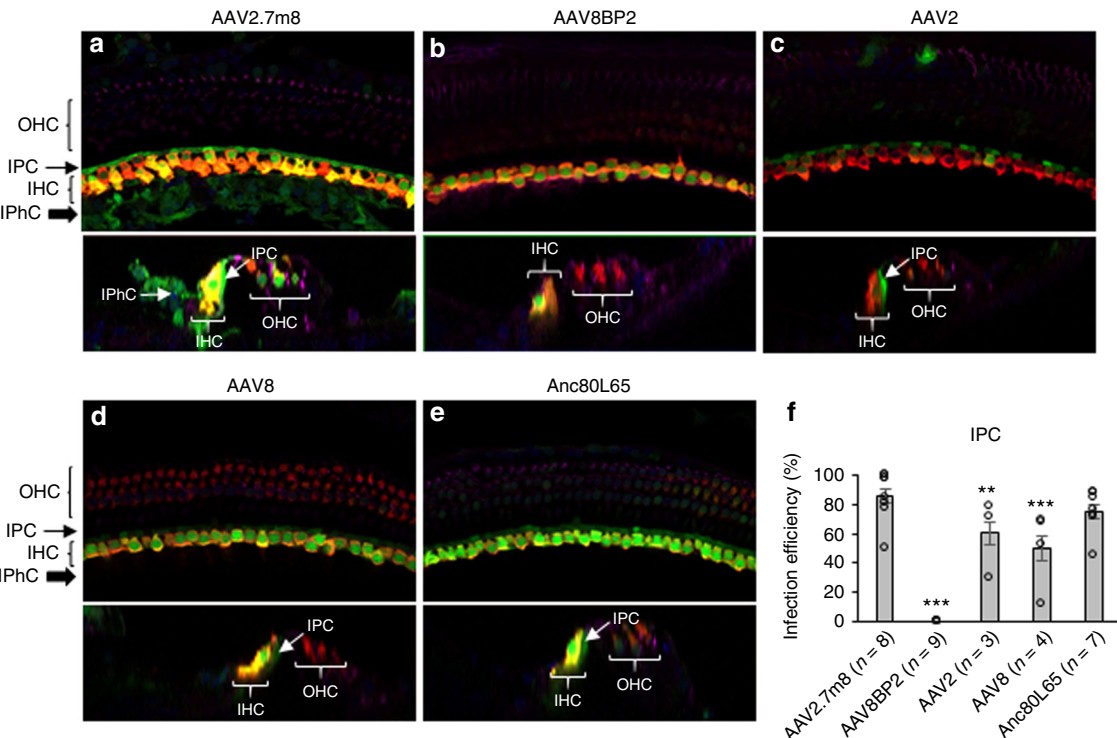

**Fig. 4** AAV2.7m8 infects inner pillar cells and inner phalangeal cells with high efficiency. **a–e** Confocal images of cochlear apex showing inner pillar cell (IPC) and inner phalangeal cell (IPhC) infection efficiency in response to posterior canal AAV delivery. AAV2.7m8-GFP (**a**) infects the IPCs and IPhCs at high levels. In contrast, AAV8BP2 (**b**) does not infect IPCs and IPhCs. AAV2-GFP (**c**), AAV8-GFP (**d**) and Anc80L65-GFP (**e**) infect the IPCs at lower levels but do not infect IPhCs. **f** Quantification of IPC infection efficiency. Statistical significance with reference to AAV2.7m8 is shown above error bars (* represents $p < 0.05$, ** represents $p < 0.01$, and *** represents $p < .001$; $t$-test). Error bars represent standard errors. Open circles represent the average infection efficiency of each animal. $n$ represents the number of animals tested. For each animal, inner pillar cell infection was quantified at six different locations along the cochlea: two at the apex, two at the middle turn, and two at the cochlear base. Source Data are provided as a Source Data file. $p$, $p$ values

that have been shown by others to be LGR5 + (inner pillar cells and inner phalangeal cells) with very high efficiency. Taken together, our results demonstrate that AAV2.7m8 is a powerful viral vector, and it will likely greatly expand the potential applications for inner ear gene therapy.

## Methods

**AAV vector construction**. The AAV2.7m8-CAG-eGFP ($9.75 \times 10^{12}$ GC/mL), AAV8BP2-CAG-eGFP ($1.10 \times 10^{13}$ GC/mL), AAV2-CAG-eGFP ($5.69 \times 10^{12}$ GC/mL), AAV2/8-CAG-eGFP ($1.66 \times 10^{13}$ GC/mL), and Anc80L65-CAG-eGFP ($1.89 \times 10^{13}$ GC/mL) were produced by the Research Vector Core at the Center for Advanced Retinal and Ocular Therapeutics (University of Pennsylvania). The production method for these viruses have been previously described[26]. All viruses were produced using the same transgene construct, consisting of the CAG promoter derived from InvivoGen pDRIVE CAG plasmid (InvivoGen, San Diego, CA, USA), the cDNA encoding enhanced GFP (eGFP) protein, and the bovine growth hormone polyadenylation signal.

**Animal surgery**. Animal surgery was approved by the Animal Care and Use Committee at the National Institute on Deafness and Other Communication Disorders (NIDCD ASP1378–18). All animal procedures were done in compliance with the ethical guidelines and regulations set forth by the Animal Care and Use Committee at the National Institute on Deafness and Other Communication Disorders. CBA/J mice were used in this study. For neonatal mice (P0–P5), hypothermia was used to induce and maintain anesthesia. Surgery was performed only in the left ear of each animal. The right ear served as a control. For inner ear gene delivery via the posterior semicircular canal approach, a post-auricular incision was made, and the tissue was dissected to expose the posterior semicircular canal. Care was taken to avoid the facial nerve during the dissection. A Nanoliter Microinjection System (Nanoliter2000; World Precision Instruments, Sarasota, FL, USA) was used in conjunction with a glass micropipette to load AAV-eGFP into the glass micropipette. A total of 1 μL of AAV-eGFP was injected over approximately 40 s. The incision was closed with 5–0 vicryl sutures.

For adult mice, anesthesia was induced using isoflurane gas (Baxter, Deerfield, IL, USA) through a nose cone at a flow rate of 0.5 L/min. The adult mouse otic capsule was completely ossified (in contrast to the neonatal mouse otic capsule, which is cartilaginous). Therefore, the adult mouse inner ear gene delivery was done using the round window approach[13,27,28]. A post-auricular incision was made using small scissors. The soft tissues were bluntly dissected to expose the bulla. A small hole was created in the bulla with a 25-gauge needle and enlarged with forceps to expose the round window (RW) membrane. A Nanoliter Microinjection System (Nanoliter2000; World Precision Instruments, Sarasota, FL, USA) was used in conjunction with a glass micropipette to load AAV-eGFP into the glass micropipette. A total of 2 μL of AAV2.7m8-eGFP ($9.75 \times 10^{12}$ GC/mL) was injected over approximately 80 s. The incision was closed with 5–0 vicryl sutures.

**Auditory brainstem response**. ABR testing was used to evaluate hearing sensitivity at ~P30. Animals were anesthetized with ketamine (100 mg/kg) and xylazine (10 mg/kg) via intraperitoneal injections and placed on a warming pad inside a sound booth (ETS-Lindgren Acoustic Systems, Cedar Park, TX, USA). The animal's temperature was maintained using a closed feedback loop and monitored using a rectal probe (CWE Incorporated, TC-1000, Ardmore, PN, USA). Sub-dermal needle electrodes were inserted at the vertex ( + ) and test-ear mastoid (-) with a ground electrode under the contralateral ear. Stimulus generation and ABR recordings were completed using Tucker Davis Technologies hardware (RZ6 Multi I/O Processor; Tucker-Davis Technologies, Gainesville, FL, USA) and software (BioSigRx, v.5.1). ABR thresholds were measured at 4, 8, 16, and 32 kHz using 3-ms, Blackman-gated tone pips presented at 29.9/s with alternating stimulus polarity. At each stimulus level, 512–1024 responses were averaged. Thresholds were determined by visual inspection of the waveforms and were defined as the lowest stimulus level at which any wave could be reliably detected. A minimum of two waveforms were obtained at the threshold level to ensure repeatability of the response. Physiological results were analyzed for individual frequencies and then averaged for each of these frequencies from 4 to 32 kHz.

**Circling behavior**. The circling behavior of mice that underwent inner ear gene delivery was quantified using optical tracking and the ANY-maze tracking software (version 4.96; Stoelting Co., Wood Dale, IL, USA). A 38× 58 cm box was attached

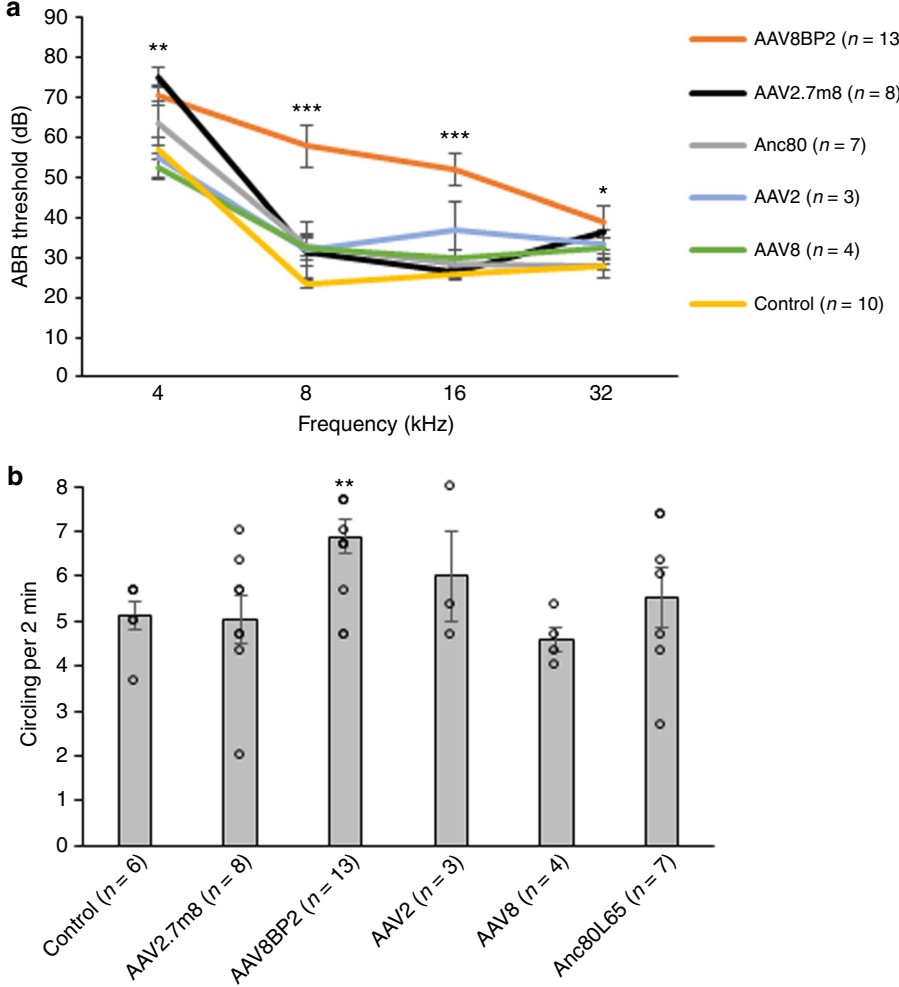

**Fig. 5** AAV2.7m8 has a minimal adverse effect on auditory and vestibular functions in injected mice. **a** Auditory brainstem responses (ABRs) were recorded to assess the auditory function of mice that underwent synthetic AAV injection via the posterior semicircular canal approach. AAV2.7m8, AAV2, AAV8, and Anc80L65 had minimal adverse effects on the auditory function, while the injection of AAV8BP2 caused a 10–25 dB ABR threshold hold elevation compared to non-injected control mice. **b** Circling behavior was assessed in mice that underwent AAV injection via the posterior semicircular canal approach. AAV2.7m8, AAV2, AAV8, and Anc80L65 did not cause statistically significant increase in circling behavior compared to non-injected control mice, while the injection of AAV8BP2 caused a slight elevation in circling behavior compared to non-injected control mice. Statistical significance with reference to non-injected normal control mice is shown above error bars (* represents $p < 0.05$, ** represents $p < 0.01$, and *** represents $p < 0.001$; ANOVA). Error bars represent standard errors. Open circles represent the average value of each animal. $n$ represents the number of animals tested. Source data are provided as a Source Data file. $p$, $p$ values

to a video camera (Fujinon YV5X2.7R4B-2 1/3-inch 2.7–13.5 mm F1.3 Day/Night Aspherical Vari-focal Lens). The ANY-maze video tracking software was set to track the head of the mice placed within the box. Each mouse was placed into the box and allowed to acclimate to the new environment for 2 min. Complete rotations were recorded and quantified for the next 2 min, followed by a 1-minute "cool-down" period where rotations were not tracked. Each mouse was assessed three times, and the average was taken.

**Immunohistochemistry and quantification.** After completion of functional testing, mice were euthanized by $CO_2$ asphyxiation followed by decapitation. Temporal bones were harvested and fixed overnight with 4% paraformaldehyde followed by decalcification in 120 mM ethylenediaminetetraacetic acid for 4 days. The vestibular organs and cochlear sensory epithelia were micro-dissected, blocked, and labeled with mouse anti-myosin 7a antibody to label hair cells (1:200, product # 25–6790; Proteus BioSciences, Ramona, CA, USA), with mouse anti-acetylated tubulin antibody to label supporting cells (1:100, product # T9026; Sigma-Aldrich Corp., St. Louis, MO, USA), and chicken anti-GFP antibody (1:1000, product # ab13970; Abcam, Cambridge, MA, USA), and Hoechst stain (1:500, product # 62249; Life Technologies, Carlsbad, CA, USA) to label nuclei. Primary and secondary antibodies were diluted in phosphate-buffered saline. Images were obtained using a Zeiss LSM780 confocal microscope at 10× and 40× using z-stacks.

For hematoxylin and eosin staining, tissues were first treated with a sucrose gradient (10–30% in phosphate-buffered saline) and then were treated with a

mixture of sucrose and embedding medium SCEM (Section-Lab Co Ltd, Japan). After freezing in liquid nitrogen, tissues were then sectioned at 10 μm thickness and hematoxylin and eosin staining was done using the Hematoxylin & Eosin Stain Kit following the manufacturer's instructions (Vector Laboratories, Inc., Burlingame, CA, USA).

For the quantification of cochlear hair cell and supporting cell infection efficiency, two 40× images were taken at the apex, middle turn, and base of cochlea. The number of hair cells and supporting with GFP expression was counted and averaged at each location along the cochlea. Each 40× image contains ~30 IHCs and ~90 OHCs. The overall infection rate was calculated by averaging the infection rates obtained from the entire cochlea. For the quantification of utricular hair cell infection efficiency, two 40× images (each containing ~300 vestibular hair cells) were taken per utricle specimen and the number of hair cells with GFP expression was counted and averaged.

**Statistics.** Student's $t$ test (two-tailed) was used to assess the differences in infection efficiency. It has been shown that different AAV serotypes can have different infection efficiencies in different regions of the cochlea[29]. Therefore, infection efficiencies from each region of the cochlea (apex, middle turn, and cochlear base) were treated as separate measurements in the calculation of mean, standard error, and statistical significance. ANOVA was used to assess the differences in ABR thresholds as well as the circling behavior. Post-hoc analysis was performed using Scheffe's method. A $p$ value < 0.05 indicates statistical significance.

**Reporting summary**. Further information on experimental design is available in the Nature Research Reporting Summary linked to this article.

## Data availability

The Source Data underlying Figs. 1, 3, 4, 5 and Supplementary Figs. 2 and 4 are provided as a Source Data file. All other data are available from the authors upon reasonable request.

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

## Acknowledgements

This work is supported by the NIDCD Division of Intramural Research grant DC000082–02 (W.W.C.), Foundation Fighting Blindness-sponsored CHOP-Penn Pediatric Center for Retinal Degenerations (J.B.), National Eye Institute/NIH grants R21EY020662 and 8DP1EY023177 (J.B.), Research to Prevent Blindness (J.B.), the Paul and Evanina Mackall Foundation Trust (J.B.), the Center for Advanced Retinal and Ocular Therapeutics (J.B.), and the F.M. Kirby Foundation (J.B.). We are grateful to the NIDCD animal facility staff for caring for our animals. We are also grateful to Shangzhen Zhou for the assistance in generating the recombinant AAVs. We also thank Dr. Lisa Cunningham and Dr. Thomas Friedman for reviewing the manuscript.

## Author contributions

W.C., D.S.M., and J.B. conceived and designed the study. K.I., J.Z. and W.C. performed and analyzed the experiments. W.C. supervised the work. H.J.W. provided support for the study. W.C. and K.I. wrote the manuscript with the participation of all the authors.

## Additional information

**Competing interests:** J.B. is a co-author on a patent for AAV8BP2; Enhanced AAV-mediated gene transfer for retinal therapies (9,567,376); Publication date: February 14, 2017; Co-inventors: Therese Cronin, Jean Bennett, Luk E. Vandenberghe. This patent describes the sequence used to generate the virus and the ability to use this capsid to more efficiently target retinal cells. W.C. and J.B. are co-authors on a pending patent, filed by the National Institutes of Health; U.S. Patent Application No. 62/784,306; for the use of AAV2.7m8 for ear applications. The remaining authors declare no competing interests.

