## [Peer Review File · Nature Communications]

Reviewers' comments:

Reviewer #1 (Remarks to the Author):

The manuscript by Isgrig et al., examines the infection patterns of three synthetic AAVs (AAV2.7m8, AAV8BP2, and AAV-DJ) in the wild-type mouse inner ear. The authors claim that AAV2.7m8 infects both inner hair cells (IHCs) and outer hair cells (OHCs) with high efficiency outperforming the other two vectors. In addition, they state that AAV2.7m8 infects inner pillar cells and inner phalangeal cells. They do not report adverse effects on inner ear physiology due to viral transduction using AAV2.7m8 confirming its promise for future therapies.

These findings are novel and will be of interest to the investigators working on inner ear gene therapies. However, the study is largely descriptive without mechanistic insight, thus limiting its interest to a wider field. Overall, the work is convincing and it provides the level of detail a researcher needs to reproduce the work. That being said, it lacks an important side-by-side comparison with a synthetic vector that has been shown to outperform naturally occurring AAVs in the inner ear (Anc80L65) giving rise to inner and outer hair cell transduction. Another limitation of this work is that vectors used for the side-by-side comparisons were purchased from different vector cores (AAV2.7m8 and AAV8BP2 were from UPenn Vector Core and AAV-DJ from Vector Labs). Different cores use different viral production, purification and titration protocols making it difficult to compare batches produced across facilities.

Below are some additional questions concerning the manuscript.

- High standard deviation was observed with AAV2.7m8 potentially due to injection into the perilymph instead of endolymph. Are there any methods to monitor where the injection was actually made prior to analysing the histology?
- What could be accountable for differential infection properties of the various artificial and natural AAVs? Are there any known differences in proteoglycans that act as viral receptors found on inner versus outer hair cells, or in the pillar/phalangeal cells that could explain and help predict the behaviour of different AAVs in the inner ear?
- The authors observe an increased ABR threshold after AAV8BP2 transduction. Can they propose a hypothesis why this occurs and only with this AAV?
- There is no mention of how many particles of each preparation was injected: there is a statement about volume of injection (about 1 ul) but did all solutions contain the same number of particles?
- The figures are very low resolution in the pdf for review: especially in the figure 3 it was difficult to see the inner pillar and inner phalangeal cells.

Reviewer #2 (Remarks to the Author):

The ear is an important, and recently emerging, target for AAV gene therapy. However, a vector capable of broad transduction of both inner and outer hair cells, vestibular hair cells, and supporting glia has not yet emerged. This study investigated the transduction properties of three engineered AAVs in the ear, and it found that 7m8 - a variant generated by directed evolution for

transduction of the outer murine retina upon intravitreal administration - shows strong transduction, particularly of cochlear hair cells and LGR5+ supporting cells. The level of transduction (>80%) was high, the study was well designed, the manuscript is well-written, and the work will thus be of interest to the field.

Several questions emerge. First, virus was administered to neonatal mice. Since gene therapy for deafness may occur at a range of ages, depending on age of onset of disease and safety considerations, it is worth considering what happens in adults. This is especially the case since the Discussion proposes age-associated hearing loss as a target of strong interest to the authors.

In addition, the study very effectively characterizes transduction, and based on these results and the lack of toxicity, one might anticipate that the vector would perform well in a model of hearing loss. However, efficacy wasn't explored in this manuscript.

Finally, examples of differences in transduction of different species - such as mouse and primate - are legion in the field. While a study in non-human primate is beyond the scope of this manuscript, it should be stated more prominently in the abstract and introduction that the results are in mouse.

Reviewer #3 (Remarks to the Author):

Most naturally occurring AAVs have shown to infect inner hair cells (IHC), but the transduction levels of outer hair cells (OHC) and supporting cells is very low. In this manuscript, the authors compared three synthetic AAVs (AAV2.7m8, AAV8BP2 and AAV-DJ) in the mouse inner ear. They showed that AAV2.7m8 is able to transduce with high efficiency IHC, OHC and supporting cells. Although the data are interesting, and have the potential of expanding the application of these serotypes for ear gene therapy, the study is not well designed. It seems more like preliminary data lacking the proper controls to really gauge the relevance of the data.

Some specific major points:

1. The study is incomplete; the authors did not include any natural occurring AAV previously used in other inner ear studies (AAV1, AAV2 and AAV5) as control. No comparison can be established with previously published data and the authors do not provide any explanation why the study was done without them.
2. There is not mention in the manuscript about the dose injected in the inner ear. In addition, the authors used different promoters for each serotype (CAG for AAV2.7m8 and AAV8bp2 versus CMV for AAV-DJ), and there is not sufficient information about the expression cassettes used to produce the three different vectors. Since dose, promoters and cassette design have showed to impact the tropism and expression levels in the inner ear and other tissues (Stone et al., 2005; Liu et al., 2007; Xiong et al., 2015) the authors should provide detailed information in

the paper. In addition, the same expression cassette should be used for the three different serotypes if the authors want to make any comparisons between them.

3. The authors stated (line 176) "Our results suggest that AAV2.7m8 is also capable of infecting cochlear IHCs and OHCs with high efficiency, comparable to that of the Anc80L65". This statement is just speculation, since this synthetic serotype was not included. Anc80L65 should have been included if comparison were to be done between these serotypes. The quantification results cannot be compared with previously published data (Landegger et al., 2016) since the injection method (round window versus semicircular canal), transgene cassette (CMV versus CAG promoter) and immunohistochemistry method (direct immunofluorescent versus staining with anti-GFP antibody) were different.

4. Also in line 107: "Some IHC loss was seen in mice with high levels of GFP expression after AAV8BP2 injection". Does GFP have a toxic effect? This is surprising since we don't see the same effect with AAV2.7m8 and GFP expression is even higher. In addition, these mice showed a significant ABR threshold elevation. Do the authors have an explanation for this results? Is this related with the serotype? Did the authors check for any inflammatory or immune response in these mice?

5. In line 198, the authors claimed "this is the first report of AAV capable of targeting the LGR5+ supporting cells with high efficiency in vivo". This statement is incorrect and inaccurate. Based on previously published data (Shi et al., 2012), LGR5+ is expressed in a very small pool of supporting cells, and is restricted to a few types of supporting cells. The authors' just use a general marker for all supporting cells (acetylated tubulin), and no additional characterization of the GFP positive supporting cells is done. So is unknown if the cells stained with GFP are LGR5+ cells.

Minor points:

1. Authors should include a more detailed description in the methods of the quantification of hair cells and supporting cells

2. It is unclear how many animals were use for each experiment (staining, cell quantification). The authors should state the number of animals on each figure.

3. The number of animals per group is different between serotypes (n=9 for AAV8BP2 and n=6 for AAV-DJ). Is there a reason for these differences?

4. Please review the error bars in Figure 1, they don't seem to match the data provided in the paper.

5. Are the exposure times the same for all the images taken? In Figure 1, the fluorescent signal for Myo7a seems lower for AAV8BP2 than for AAV2.7m8. Please review.

6. In figure 2, the morphology of the utricle in A looks different from B and C. Please change, so the reviewer can compare the fluorescent signal between them. In addition review error bars in the graph.

7. Please increase the size of the pictures in Figure S1, so the staining can be examined properly

Response to reviewers

Dear Editor and Reviewers,

We sincerely appreciate your thoughtful suggestions on our manuscript. The following is a list of responses to your comments.

Reviewers' comments:

Reviewer #1 (Remarks to the Author):

The manuscript by Isgrig et al., examines the infection patterns of three synthetic AAVs (AAV2.7m8, AAV8BP2, and AAV-DJ) in the wild-type mouse inner ear. The authors claim that AAV2.7m8 infects both inner hair cells (IHCs) and outer hair cells (OHCs) with high efficiency outperforming the other two vectors. In addition, they state that AAV2.7m8 infects inner pillar cells and inner phalangeal cells. They do not report adverse effects on inner ear physiology due to viral transduction using AAV2.7m8 confirming its promise for future therapies.

These findings are novel and will be of interest to the investigators working on inner ear gene therapies. However, the study is largely descriptive without mechanistic insight, thus limiting its interest to a wider field. Overall, the work is convincing and it provides the level of detail a researcher needs to reproduce the work. That being said, it lacks an important side-by-side comparison with a synthetic vector that has been shown to outperform naturally occurring AAVs in the inner ear (Anc80L65) giving rise to inner and outer hair cell transduction.

Response: We have added transduction data with Anc80L65 as well as conventional AAVs (AAV2 and AAV8) in the revised manuscript. Our data indicate that AAV2.7m8 is comparable to Anc80L65 at transducing IHCs, and superior at transducing OHCs. In addition, AAV2.7m8 is much more effective at transducing the inner phalangeal cells compared to Anc80L65.

Another limitation of this work is that vectors used for the side-by-side comparisons were purchased from different vector cores (AAV2.7m8 and AAV8BP2 were from UPenn Vector Core and AAV-DJ from Vector Labs). Different cores use different viral production, purification and titration protocols making it difficult to compare batches produced across facilities.

Response: While it would be more ideal to always obtain all viruses from a single source, it is not always possible. This is not unusual in published inner ear gene therapy studies (e.g. Askew et al., *Science Translational Medicine* 2015; Shu et al., *Human Gene Therapy* 2016). We have pointed out this limitation in the revised manuscript:

Page 5, lines 103-106: "It is important to note AAV-DJ-GFP was obtained from a different source (Vector Biolabs) compared to all other AAVs used in this study (UPenn). This could potentially contribute to the low levels of hair cell infection observed with AAV-DJ-GFP."

Below are some additional questions concerning the manuscript.

- High standard deviation was observed with AAV2.7m8 potentially due to injection into the perilymph

instead of endolymph. Are there any methods to monitor where the injection was actually made prior to analyzing the histology?

Response: As the reviewer pointed out, we suspect that the variability seen with some of our data may come from injection into perilymph vs. endolymph. Unfortunately it is not possible to assess whether injection is in perilymph or endolymph due to the small size of the mouse labyrinth. While some investigators have reported excluding data from animals with suspected perilymph injections, we decided to report all of our data to give the readers a sense of the variability that is sometimes encountered during these experiments.

-What could be accountable for differential infection properties of the various artificial and natural AAVs? Are there any known differences in proteoglycans that act as viral receptors found on inner versus outer hair cells, or in the pillar/phalangeal cells that could explain and help predict the behaviour or different AAVs in the inner ear?

Response: The main receptor for AAV2 is heparan sulfate proteoglycan. The capsid of AAV2.7m8 has modifications to the heparan sulfate binding site which are thought to enhance its ability to transduce target cells. This is included in the revised text:

Page 4, lines 69-74: "AAV2.7m8 was generated using an *in vivo*-directed evolution approach where AAV libraries with diverse capsid protein modifications were screened for infection efficiency of mouse photoreceptor cells via intravitreal injection. This vector contains a 10-amino acid peptide inserted at position 588 of the AAV2 capsid protein sequence, which is involved with AAV2 binding to its primary receptor, heparan sulfate proteoglycan."

- The authors observe an increased ABR threshold after AAV8BP2 transduction. Can they propose a hypothesis why this occurs and only with this AAV?

Response: We suspect that AAV8BP2 may trigger a more robust inflammatory response in the cochlea compared to other AAVs used in the study. Examination of the cochlea after AAV8BP2 injection showed infiltration of inflammatory cells on H&E stain. When the concentration of AAV8BP2 was reduced in half (0.5×10^{10} G.C.), we found that it did not cause ABR threshold elevation, but the hair cell transduction efficiency was lower. The proposed hypothesis and additional data are now included in the revised text as follows:

Pages 8-9, lines 180-188: "It is possible that AAV8BP2 is more immunogenic to the mouse inner ear, which leads to cochlear hair cell loss (**Fig. 1**) as well as ABR threshold elevation. Examination of the cochlea after AAV8BP2 injection revealed infiltration of inflammatory cells (**Suppl. Fig. 2**). When AAV8BP2-GFP was injected at half of the original concentration (0.5×10^{10} G.C.), the ABR thresholds were comparable to control mice ($p=0.49$, **Suppl. Fig. 3**), but the IHC and OHC infection efficiency also decreased ($43.2 \pm 8.36\%$ and $23.3 \pm 5.41\%$, respectively, $n=5$), though the changes were not statistically significant ($p=0.38$ and 0.08 for IHC and OHC respectively). "

- There is no mention of how many particles of each preparation was injected: there is a statement about volume of injection (about 1 ul) but did all solutions contain the same number of particles?

Response: Approximately 1×10^{10} viral particles were injected for each virus. This is now included in the revised methods:

Pages 4-5, lines 90-91: "Approximately 1×10^{10} genome copies (G.C.) were delivered into the inner ear of each animal."

- The figures are very low resolution in the pdf for review: especially in the figure 3 it was difficult to see the inner pillar and inner phalangeal cells.

Response: We have included additional higher resolution images of the inner pillar and inner phalangeal cell transduction in the revised manuscript (Figure 4 and Supplementary Figure 1).

Reviewer #2 (Remarks to the Author):

The ear is an important, and recently emerging, target for AAV gene therapy. However, a vector capable of broad transduction of both inner and outer hair cells, vestibular hair cells, and supporting glia has not yet emerged. This study investigated the transduction properties of three engineered AAVs in the ear, and it found that 7m8 - a variant generated by directed evolution for transduction of the outer murine retina upon intravitreal administration - shows strong transduction, particularly of cochlear hair cells and LGR5+ supporting cells. The level of transduction (>80%) was high, the study was well designed, the manuscript is well-written, and the work will thus be of interest to the field.

Several questions emerge. First, virus was administered to neonatal mice. Since gene therapy for deafness may occur at a range of ages, depending on age of onset of disease and safety considerations, it is worth considering what happens in adults. This is especially the case since the Discussion proposes age-associated hearing loss as a target of strong interest to the authors.

Response: We agree with the reviewer that the infection patterns in adult mouse cochlea is an important topic. In fact, we are in the process of completing a separate study focusing on the infection patterns of synthetic AAVs in adult mouse cochlea. Our preliminary data suggest that AAV2.7m8 infects both the IHCs and OHCs at very high efficiency, similar to what was observed in neonatal mouse cochlea. However, it is our opinion that the adult cochlear infection data is beyond the scope of this study, which focuses on the synthetic AAV infection patterns in neonatal mouse inner ear.

In addition, the study very effectively characterizes transduction, and based on these results and the lack of toxicity, one might anticipate that the vector would perform well in a model of hearing loss. However, efficacy wasn't explored in this manuscript.

Response: We are working on several mouse models of hereditary hearing loss using the AAV2.7m8 virus as the viral vector. Since these studies require extensive characterization of the animal models, we plan to submit these data as separate publications. It is worthwhile to point out that in the study by Landegger et al. in Nature Biotechnology 2017, where the transduction properties of Anc80L65 in neonatal mouse inner ear were reported, no animal model of hearing loss was included.

Finally, examples of differences in transduction of different species - such as mouse and primate - are

legion in the field. While a study in non-human primate is beyond the scope of this manuscript, it should be stated more prominently in the abstract and introduction that the results are in mouse.

Response: We are planning to test the transduction properties of AAV2.7m8 in human vestibular tissues. We have added the language below in the revised manuscript to point out this limitation in our study:

Page 10, lines 215-218: "While our data indicate AAV2.7m8 is highly effective at infection of cochlear hair cells in the mouse cochlea, it remains important to confirm these findings in larger animals such as non-human primates in order for this virus to be useful in human inner ear gene therapy."

Reviewer #3 (Remarks to the Author):

Most naturally occurring AAVs have shown to infect inner hair cells (IHC), but the transduction levels of outer hair cells (OHC) and supporting cells is very low. In this manuscript, the authors compared three synthetic AAVs (AAV2.7m8, AAV8BP2 and AAV-DJ) in the mouse inner ear. They showed that AAV2.7m8 is able to transduce with high efficiency IHC, OHC and supporting cells. Although the data are interesting, and have the potential of expanding the application of these serotypes for ear gene therapy, the study is not well designed. It seems more like preliminary data lacking the proper controls to really gauge the relevance of the data.

Some specific major points:

1. The study is incomplete; the authors did not include any natural occurring AAV previously used in other inner ear studies (AAV1, AAV2 and AAV5) as control. No comparison can be established with previously published data and the authors do not provide any explanation why the study was done without them.

Response: We have added transduction data for AAV2, AAV8, and Anc80L65 to the revised manuscript. We chose AAV2 and AAV8 for two reasons: 1) both AAVs are commonly used in animal and human gene therapy studies, and 2) AAV2.7m8 and AAV8BP2 are derived from AAV2 and AAV8, respectively.

2. There is not mention in the manuscript about the dose injected in the inner ear. In addition, the authors used different promoters for each serotype (CAG for AAV2.7m8 and AAV8bp2 versus CMV for AAV-DJ), and there is not sufficient information about the expression cassettes used to produce the three different vectors. Since dose, promoters and cassette design have showed to impact the tropism and expression levels in the inner ear and other tissues (Stone et al., 2005; Liu et al., 2007; Xiong et al., 2015) the authors should provide detailed information in the paper. In addition, the same expression cassette should be used for the three different serotypes if the authors want to make any comparisons between them.

Response: Approximately 1×10^{10} viral particles were injected for each virus. This is now included in the revised methods. In order to address the Reviewer's concern about differences in promoters, we obtained AAV-DJ-CAG-GFP and have added the data from this virus to the revised text in order to make

the promoter consistent among all the viruses used in the study. Relevant changes to the text are shown below:

Pages 4-5, lines 90-91: “Approximately 1×10^{10} genome copies (G.C.) were delivered into the inner ear of each animal.”

Page 14, lines 335-340: “The AAV2.7m8-CAG-eGFP, AAV8BP2-CAG-eGFP, AAV2-CAG-eGFP, AAV2/8-CAG-eGFP, and Anc80L65-CAG-eGFP were produced by the Research Vector Core at the Center for Advanced Retinal and Ocular Therapeutics (University of Pennsylvania). The production method for these viruses has been previously described³⁰. AAV-DJ-CAG-eGFP was purchased from Vector Biolabs (Malvern, PA). The concentration of viral stock solution was 1×10^{13} genome copies (G.C.) per ml for each virus.”

3. The authors stated (line 176) “Our results suggest that AAV2.7m8 is also capable of infecting cochlear IHCs and OHCs with high efficiency, comparable to that of the Anc80L65”. This statement is just speculation, since this synthetic serotype was not included. Anc80L65 should have been included if comparison were to be done between these serotypes. The quantification results cannot be compared with previously published data (Landegger et al., 2016) since the injection method (round window versus semicircular canal), transgene cassette (CMV versus CAG promoter) and immunohistochemistry method (direct immunofluorescent versus staining with anti-GFP antibody) were different.

Response: We have added transduction data with Anc80L65 as well as conventional AAVs in the revised manuscript. Our data indicate that AAV2.7m8 is comparable to Anc80L65 at transducing IHCs, and superior at transducing OHCs. In addition, AAV2.7m8 is much more effective at transducing the inner phalangeal cells compared to Anc80L65.

4. Also in line 107: “Some IHC loss was seen in mice with high levels of GFP expression after AAV8BP2 injection”. Does GFP have a toxic effect? This is surprising since we don’t see the same effect with AAV2.7m8 and GFP expression is even higher. In addition, these mice showed a significant ABR threshold elevation. Do the authors have an explanation for this results? Is this related with the serotype? Did the authors check for any inflammatory or immune response in these mice?

Response: We have removed the language about high levels of GFP causing ototoxicity in the revised manuscript. We suspect that AAV8BP2 may trigger a more robust inflammatory response in the cochlea compared to other AAVs used in the study. Examination of the cochlea after AAV8BP2 injection showed infiltration of inflammatory cells on H&E stain. When the concentration of AAV8BP2 was halved (0.5×10^{10} G.C.), we found that it did not cause ABR threshold elevation, but the hair cell transduction efficiency was lower. The proposed hypothesis and additional data are now included in the revised text as follows:

Pages 8-9, lines 180-188: “It is possible that AAV8BP2 is more immunogenic to the mouse inner ear, which leads to cochlear hair cell loss (**Fig. 1**) as well as ABR threshold elevation. Examination of the cochlea after AAV8BP2 injection revealed infiltration of inflammatory cells (**Suppl. Fig. 2**). When AAV8BP2-GFP was injected at half of the original concentration (0.5×10^{10} G.C.), the ABR thresholds were comparable to control mice ($p=0.49$, **Suppl. Fig. 3**), but the IHC and OHC infection efficiency also decreased ($43.2 \pm 8.36\%$ and $23.3 \pm 5.41\%$, respectively, $n=5$), though the changes were not statistically significant ($p=0.38$ and 0.08 for IHC and OHC respectively). “

5. In line 198, the authors claimed “this is the first report of AAV capable of targeting the LGR5+ supporting cells with high efficiency in vivo”. This statement is incorrect and inaccurate. Based on previously published data (Shi et al., 2012), LGR5+ is expressed in a very small pool of supporting cells, and is restricted to a few types of supporting cells. The authors’ just use a general marker for all supporting cells (acetylated tubulin), and no additional characterization of the GFP positive supporting cells is done. So is unknown if the cells stained with GFP are LGR5+ cells.

Response: The location of inner pillar and inner phalangeal cells are very characteristic and consistent. Therefore, we would suggest that our data is sufficient to show that these two supporting cell types are efficiently transduced by AAV2.7m8. These two cell types have also been shown to be LGR5+ cells in the mammalian inner ear (Shi et al., 2012, Chai et al., 2012). We have changed the language in the revised text to soften the claim about LGR5+ cells according to the reviewer’s suggestion. Revised text is as follows:

Page 11, lines 237-242: “In this study, we showed that AAV2.7m8 effectively infects both the inner and outer hair cells in the cochlea. In addition, it also infects the types of supporting cells that have been shown by others to be LGR5+ (inner pillar cells and inner phalangeal cells) with very high efficiency. Taken together, our results demonstrate that AAV2.7m8 is a powerful viral vector, and it will likely greatly expand the potential applications for inner ear gene therapy.”

Minor points:

1. Authors should include a more detailed description in the methods of the quantification of hair cells and supporting cells

Response: We have revised the manuscript according to the reviewer’s suggestion. Revised text is as follows:

Page 17, lines 399-406: “For quantification of cochlear hair cell and supporting cell infection efficiency, two 40x images were taken at the apex, middle turn, and base of cochlea. The number of hair cells and supporting cells with GFP expression was counted and averaged at each location along the cochlea. Each 40x image contains ~30 IHCs and ~90 OHCs. The overall infection rate was calculated by averaging the infection rates obtained from the entire cochlea. For quantification of utricular hair cell infection efficiency, two 40x images (each containing ~300 hair cells) were taken per utricle specimen and the number of hair cells with GFP expression was counted and averaged.”

2. It is unclear how many animals were used for each experiment (staining, cell quantification). The authors should state the number of animals on each figure.

Response: We have revised the manuscript according to the reviewer’s suggestion. The number of animals used in all experiments are now clearly shown both in the text as well as in the figures.

3. The number of animals per group is different between serotypes (n=9 for AAV8BP2 and n=6 for AAV-DJ). Is there a reason for these differences?

Response: We typically test each virus using a separate mouse litter. Since the litter size can vary, the number of pups used for each experiment may also differ. In addition, some pups are cannibalized by the mother after gene therapy delivery. This variation in the number of animals used is common in inner ear gene therapy literature.

4. Please review the error bars in Figure 1, they don't seem to match the data provided in the paper.

Response: We have revised the figure and corrected the error bars.

5. Are the exposure times the same for all the images taken? In Figure 1, the fluorescent signal for Myo7a seems lower for AAV8BP2 than for AAV2.7m8. Please review.

Response: We have confirmed that the exposure time is the same for all images taken.

6. In figure 2, the morphology of the utricle in A looks different from B and C. Please change, so the reviewer can compare the fluorescent signal between them. In addition review error bars in the graph.

Response: We have revised figure 2 to according to the reviewer's suggestions.

7. Please increase the size of the pictures in Figure S1, so the staining can be examined properly

Response: We have increased the size of Figure S1 (now Figure 2) according to the reviewer's suggestion.

Reviewers' comments:

Reviewer #1 (Remarks to the Author):

The manuscript has been improved with the inclusion on the data concerning the additional serotypes as well as the inclusion of further details and arguments in the text.

Reviewer #2 (Remarks to the Author):

The authors have addressed the prior concerns of this reviewer.

Reviewer #3 (Remarks to the Author):

The authors describe a study in which various 'designer AAVs are compared for transduction efficiencies following an intracochlear injection. Cochlear gene therapy for hearing and balance disorders is an emerging field in which gene transfer technologies with appropriate tropism and efficiencies is currently rate limiting so this is a relevant area of study. The results described here are illustrating mainly the potency of a vector called AAV7m8 which is an AAV2 variant with a peptide insertion which was first developed for retinal applications.

The results demonstrate that indeed 7m8 overall is a powerful vector system that may be superior to previously proposed vectors such as Anc80 for targeting various hair cell, most notably OHC. The revisions and rebuttal of the authors only in part address the concerns of the reviewers however clear limitations to the work still remain.

- In general, the level of experimental detail is below standard and certainly below expectations for this journal. The rebuttal appears dismissive about important technical details particularly those relating to vectors, vector dosing, and potential immunogenicity. Examples are that 'approximately' $1E10$ pt was injected, the AAV DJ vector was not retitred which is a relatively simple follow up experiment following the reviewers requests, immunogenicity is noted but with little to no follow up even though this is a real concern when one proposes a translational relevance. To address these reviewer's concerns full detail has to be provided on viral stock titers, identical transgene constructs (not just similar promoter configurations), injected volumes, side-by-side analytical assays on vector lots, and evaluation of potential safety concerns. These details matter, especially since a quantitative argument is being made in terms of the relative performance of the vectors tested. The response that not all studies were previously held to this reasonable standard is not acceptable, as these relatively routine quality control steps are required for the authors to draw the conclusions they do.

- as other reviewers point out, this study is limited to the route of injection, at a new born stage, and the species tested with no data to illustrate whether 7m8 or any of the other vectors are capable of transduction via a more translationally relevant route (i.e. perilymphatic), in more mature cochleae, and in other species. This is particularly a critical question since other peptide inserted AAVs (such as php.b) were previously shown to be restricted in their beneficial tropism to a single murine species.

Response to reviewers

Dear Editor,

Thank you for sending the three reviewers' comments to our revised manuscript entitled "AAV2.7m8 is a powerful viral vector for inner ear gene therapy (NCOMMS-18-09920A)." We are pleased to learn that reviewers #1 and #2 are satisfied with our revision. In response to Reviewer 3 and your guidance, we have added data to the revised manuscript showing that AAV2.7m8 infects the adult cochlear hair cells as well as neonatal cochlear cells (overall infection efficiency $84.5 \pm 4.91\%$ for IHCs and $74.9 \pm 6.53\%$ for OHCs, $n=6$). This is shown as the new supplementary figure 1:

Supplementary Figure 1: AAV2.7m8 infects adult mouse cochlear hair cells with high efficiency.

Confocal images of the cochlear middle turn from an adult (6 month old) CBA/J mouse injected with AAV2.7m8-GFP via the round window approach. Robust GFP expression is seen in both IHCs and OHCs, indicating high infection efficiency. GFP expression is shown in green, and Myo7a expression (a marker for hair cells) is shown in red. Scale bar represents 20 μm .

In neonatal mouse inner ear, the otic capsule is cartilaginous. Therefore, the injection micropipette can penetrate directly into the posterior semicircular canal for gene delivery. In contrast, the adult otic capsule is completely ossified, and cannulation of the posterior semicircular canal requires additional drilling to remove its bony covering. Because of this, adult gene delivery was done through the round window, which is a naturally occurring membrane at the base of the cochlea that allows for direct penetration by the micropipette. We and others have shown that the round window approach is a safe and effective way for gene delivery in adult mouse inner ear¹⁻³. The round window gene delivery approach is added to the methods section of the revised text:

Pages 14, lines 337-348: For adult mice, anesthesia was induced using isoflurane gas (Baxter, Deerfield, IL) through a nose cone at a flow rate of 0.5 L/min. The adult mouse otic capsule is completely ossified (in contrast to the neonatal mouse otic capsule which is cartilaginous). Therefore, adult mouse inner ear gene delivery was done using the round window approach, as previously described¹⁻³. Briefly, a post-auricular incision was made using small scissors. The soft tissues were bluntly dissected to expose the bulla. A small hole was created in the bulla with a 25-gauge needle and enlarged with forceps to expose the RW membrane. A Nanoliter Microinjection System (Nanoliter2000, World Precision Instruments, Sarasota, FL) was used in conjunction with a glass micropipette to load AAV-eGFP into the glass micropipette. A total of 2 μl of AAV2.7m8-eGFP (9.75×10^{12} GC/mL) was injected over approximately 80 seconds. Incision was closed with 5-0 vicryl sutures.

Below please find our responses to Reviewer #3's comments:

1. "The results demonstrate that indeed 7m8 overall is a powerful vector system that may be superior to previously proposed vectors such as Anc80 for targeting various hair cell, most notably OHC. The

revisions and rebuttal of the authors only in part address the concerns of the reviewers however clear limitations to the work still remain.”

Response: We appreciate Reviewer #3’s remark that AAV2.7m8 is a powerful vector system for inner ear gene delivery. We are grateful that Reviewers #1 and #2 are both satisfied with the revised manuscript.

2. “The rebuttal appears dismissive about important technical details particularly those relating to vectors, vector dosing, and potential immunogenicity. Examples are that 'approximately' 1E10 pt was injected, the AAV DJ vector was not retitred which is a relatively simple follow up experiment following the reviewers requests, immunogenicity is noted but with little to no follow up even though this is a real concern when one proposes a translational relevance. To address these reviewer's concerns full detail has to be provided on viral stock titers, identical transgene constructs (not just similar promoter configurations), injected volumes, side-by-side analytical assays on vector lots, and evaluation of potential safety concerns. These details matter, especially since a quantitative argument is being made in terms of the relative performance of the vectors tested. The response that not all studies were previously held to this reasonable standard is not acceptable, as these relatively routine quality control steps are required for the authors to draw the conclusions they do.”

Response: We take all reviewers’ comments seriously and apologize if our rebuttal appeared to be dismissive. In the revised manuscript, we have included the exact viral titer used for each virus in both the main text as well as in the method section. We have also included a detailed description of the transgene construct in the method section. Since reviewer #3 had significant concerns about AAV-DJ not being made by the same facility as the other viruses, we have removed all AAV-DJ data from the revised text to ensure that all the viruses used in this study came from the same facility and had the exact same transgene construct. Relevant changes to the text are noted below:

Page 4, lines 78-87: To assess the infection efficiency of synthetic AAVs in the mammalian inner ear, AAV2.7m8-GFP (9.75×10^{12} genome copies (GC)/mL) and AAV8BP2-GFP (1.10×10^{13} GC/mL) were delivered to neonatal (P0-P5) mouse inner ears using the posterior semicircular canal approach. Posterior semicircular canal gene delivery allows viral vectors to effectively infect cells in the neonatal cochlea as well as vestibular organs⁴⁻⁶. Infection efficiencies of AAV2-GFP (5.69×10^{12} GC/mL) and AAV8-GFP (1.66×10^{13} GC/mL), two commonly used conventional AAVs from which AAV2.7m8 and AAV8BP2 are derived from respectively, as well as the synthetic AAV Anc80L65-GFP (1.89×10^{13} GC/mL), were also examined using the same delivery approach as additional controls.

Page 13, lines 316-324: The AAV2.7m8-CAG-eGFP (9.75×10^{12} GC/mL), AAV8BP2-CAG-eGFP (1.10×10^{13} GC/mL), AAV2-CAG-eGFP (5.69×10^{12} GC/mL), AAV2/8-CAG-eGFP (1.66×10^{13} GC/mL), and Anc80L65-CAG-eGFP (1.89×10^{13} GC/mL) were produced by the Research Vector Core at the Center for Advanced Retinal and Ocular Therapeutics (University of Pennsylvania). The production method for these viruses have been previously described⁷. All viruses were produced using the same transgene construct, consisting of the CAG promoter derived from InvivoGen pDRIVE CAG plasmid (InvivoGen, San Diego, CA), the cDNA encoding enhanced GFP (eGFP) protein, and the bovine growth hormone (bGH) polyadenylation signal.

3. “.....as other reviewers point out, this study is limited to the route of injection, at a new born stage, and the species tested with no data to illustrate whether 7m8 or any of the other vectors are capable of transduction via a more translationally relevant route (i.e. perilymphatic), in more mature cochleae, and in other species. This is particularly a critical question since other peptide inserted AAVs (such as php.b) were previously shown to be restricted in their beneficial tropism to a single murine species.”

Response: We have added data showing that AAV2.7m8 infects the adult cochlear hair cells equally well (overall infection efficiency $84.5 \pm 4.91\%$ for IHCs and $74.9 \pm 6.53\%$ for OHCs, n=6). This is shown as the new supplementary figure 1. The fact that gene delivery in adult mouse inner ear was done using the round window approach addresses Reviewer #3's concern about translationally relevant route. While there are well-established surgical approaches to access human semicircular canals (e.g. surgical repair of superior semicircular canal dehiscence, plugging of posterior semicircular canal for intractable benign positional vertigo), the round window approach is likely to be the most translationally relevant route for inner ear gene delivery in humans, since it is the least invasive method for accessing the human inner ears. The fact that AAV2.7m8 transduces the cochlear hair cells at very high efficiencies in both neonatal and adult mouse inner ears when delivered through both the posterior canal and the round window approaches suggest that it is a powerful viral vector for inner ear gene delivery.

References:

1. Chien, W.W., McDougald, D.S., Roy, S., Fitzgerald, T.S. & Cunningham, L.L. Cochlear gene transfer mediated by adeno-associated virus: Comparison of two surgical approaches. *Laryngoscope* (2015).
2. Zhu, B.Z., Saleh, J., Isgrig, K.T., Cunningham, L.L. & Chien, W.W. Hearing Loss after Round Window Surgery in Mice Is due to Middle Ear Effusion. *Audiol Neurootol* **21**, 356-364 (2016).
3. Akil, O., Rouse, S.L., Chan, D.K. & Lustig, L.R. Surgical method for virally mediated gene delivery to the mouse inner ear through the round window membrane. *J Vis Exp* (2015).
4. Isgrig, K., *et al.* Gene Therapy Restores Balance and Auditory Functions in a Mouse Model of Usher Syndrome. *Mol Ther* **25**, 780-791 (2017).
5. Suzuki, J., Hashimoto, K., Xiao, R., Vandenberghe, L.H. & Liberman, M.C. Cochlear gene therapy with ancestral AAV in adult mice: complete transduction of inner hair cells without cochlear dysfunction. *Scientific reports* **7**, 45524 (2017).
6. Tao, Y., *et al.* Delivery of Adeno-Associated Virus Vectors in Adult Mammalian Inner-Ear Cell Subtypes Without Auditory Dysfunction. *Hum Gene Ther* (2018).
7. Ramachandran, P.S., *et al.* Evaluation of Dose and Safety of AAV7m8 and AAV8BP2 in the Non-Human Primate Retina. *Hum Gene Ther* **28**, 154-167 (2017).

REVIEWERS' COMMENTS:

Reviewer #2 (Remarks to the Author):

This revision addresses several remaining concerns of reviewer 3. Specifically, the text was revised to include better characterization of the vector preps/lots/dosages. In addition, and very importantly, the authors provide new data showing strong transduction after administration (via the round window) to adult mice, which further increases impact of the work.